# Text-Infused Attention and Foreground-Aware Modeling for Zero-Shot Temporal Action Detection

**Yearang Lee**     **Ho-Joong Kim**     **Seong-Whan Lee**[*]
Dept. of Artificial Intelligence, Korea University, Seoul, Korea
{yr_lee, hojoong_kim, sw.lee}@korea.ac.kr

## Abstract

Zero-Shot Temporal Action Detection (ZSTAD) aims to classify and localize action segments in untrimmed videos for unseen action categories. Most existing ZSTAD methods utilize a foreground-based approach, limiting the integration of text and visual features due to their reliance on pre-extracted proposals. In this paper, we introduce a cross-modal ZSTAD baseline with mutual cross-attention, integrating both text and visual information throughout the detection process. Our simple approach results in superior performance compared to previous methods. Despite this improvement, we further identify a common-action bias issue that the cross-modal baseline over-focus on common sub-actions due to a lack of ability to discriminate text-related visual parts. To address this issue, we propose Text-infused attention and Foreground-aware Action Detection (Ti-FAD), which enhances the ability to focus on text-related sub-actions and distinguish relevant action segments from the background. Our extensive experiments demonstrate that Ti-FAD outperforms the state-of-the-art methods on ZSTAD benchmarks by a large margin: 41.2% (+ 11.0%) on THUMOS14 and 32.0% (+ 5.4%) on ActivityNet v1.3. Code is available at: https://github.com/YearangLee/Ti-FAD.

## 1   Introduction

Temporal action detection (TAD) is a fundamental task in video understanding, aiming to localize and classify action instances within untrimmed videos. Conventional TAD methods are time-consuming and expensive for annotating long untrimmed videos, making them impractical for real-world scenarios. Zero-shot temporal action detection (ZSTAD) [8, 18, 20, 26, 29] has been introduced to recognize unseen action categories that are not included in the training dataset by utilizing arbitrary categories defined by textual descriptions. As large-scale pre-trained visual-language (ViL) models (e.g. CLIP [21] and ALIGN [6]) have shown impressive performance on zero-shot downstream tasks, recent works [18, 20, 26] have extended their application to ZSTAD.

Most existing ZSTAD methods [8, 18, 20] employ a foreground-based approach that initially generates the foreground candidate proposals from the visual features and subsequently fuses them with the text features, as shown in Fig. 1 (a). This foreground-based approach fuses only text features with visual features in the foreground, following a previous ViL model trained on an environment containing the foreground. However, addressing only foreground features limits the use of the entire information of text-visual modalities due to the following two reasons: (1) the text features are only incorporated with the foreground candidate proposals that contain the partial information of the video, and (2) the visual features are utilized for localization without incorporating the text features.

To demonstrate the importance of incorporating both text and video information throughout the entire detection process, we construct a simple cross-modal baseline employing the attention mechanism [25], as illustrated in Fig. 1 (b). Even with this simple approach, we confirm that our cross-modal

---

[*]Corresponding author

38th Conference on Neural Information Processing Systems (NeurIPS 2024).

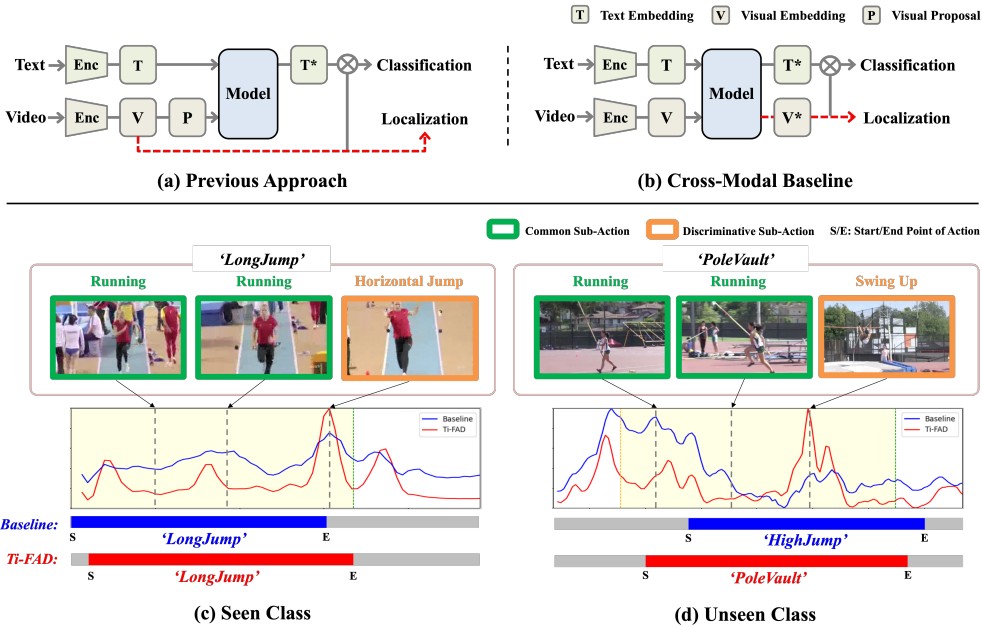

Figure 1: Top: the structure comparison of (a) the previous foreground-based approach and (b) our cross-modal baseline. Bottom: the example of the common-action bias issue. The yellow section represents ground-truth segments. The blue and red lines denote the classification scores for our cross-modal baseline and Ti-FAD, respectively.

baseline achieves superior performance compared to the previous methods [8, 18, 20] (Sec. 3.2). However, we observe an *common-action bias issue* in our cross-modal baseline that the classification score tends to capture the common sub-actions within the ground truth.

To illustrate this issue, we investigate classification scores of seen and unseen classes. Fig. 1 ((c)-(d)) shows an example of the *common-action bias issue* by comparing the classification scores that represent the similarity of text and video features. As shown in Fig. 1 (c), our cross-modal baseline (blue line) achieves focusing on the text-related parts but fails to clearly differentiate between two sub-actions *Running* and *Horizontal Jump*, resulting in the model focusing on common sub-actions regardless of the specific text input. As shown in Fig. 1 (d), for the unseen class, our baseline (blue line) over-focuses on the common sub-actions (e.g., *Running*) that are common visual information contained in the seen ground-truth set. Consequently, our cross-modal baseline incorrectly detects the *PoleVault* action as *HighJump*. *How can we enable the model to focus on the text-related discriminative sub-action (e.g., Swing Up) to alleviate the common-action bias issue?*

In this paper, we propose a novel Text-infused attention and Foreground-aware Action Detection (Ti-FAD) method, which captures discriminative sub-actions for accurate detection of action instances. Built upon our cross-modal baseline model, Ti-FAD introduces two key modules to solve the *common action bias issue*: (1) Text-infused Cross Attention (TiCA) that enables the model to focus on the discriminative sub-actions that are most relevant to the text description, and (2) a foreground-aware head that accurately distinguishes action segments from the background. These enhancements enable Ti-FAD to improve the integration of text and visual information, leading to more precise classification and localization of action instances in untrimmed videos. As shown in Fig. 1 ((c)-(d)), our Ti-FAD (red line) captures the discriminative sub-action parts in the video.

In summary, our main contributions are as follows:

- We construct a novel cross-modal baseline that integrates text and visual features throughout the entire temporal action detection process.

- We propose Ti-FAD, which incorporates TiCA and a foreground-aware head to focus on discriminative sub-actions, particularly in unseen action categories.

- Our extensive experiments on THUMOS14 and ActivityNet v1.3 demonstrate that our Ti-FAD outperforms state-of-the-art methods by a considerable margin.

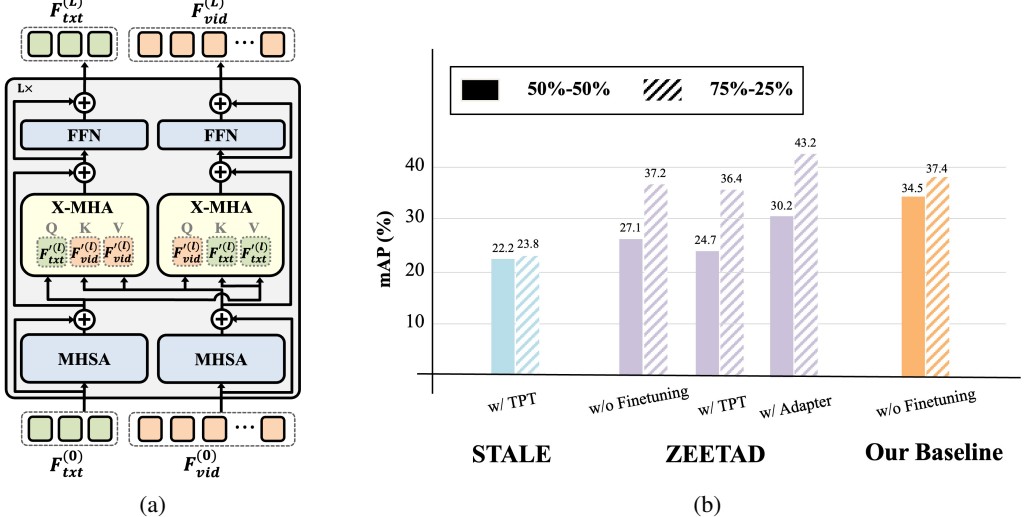

(a)                                    (b)

Figure 2: (a) Structure of our cross-modal baseline. (b) Comparison with existing methods under 50%-50% and 75%-25% settings on THUMOS14. TPT denotes text prompt tuning.

## 2 Related Work

**Vision-Language Models.** CLIP [21] and ALIGN [6] perform cross-modal contrastive learning on image-text pairs, exhibiting robust performance in open-set image classification tasks. Previous works [8, 18, 20] have explored adaptation methods such as text prompt tuning [10] to apply these models for downstream tasks. In this work, we utilize the CLIP text encoder to extract text features directly, without employing any text prompt tuning.

**Temporal Action Detection.** Existing TAD methods [3, 12, 13, 14, 23, 9, 22, 27] have explored various approaches for localizing and classifying actions in untrimmed videos. Among these methods, anchor-free detectors [12, 22, 27] have demonstrated superior performance by capturing long-length actions and simplifying the detection process. In this work, we adopt ActionFormer [27] as our baseline detector because of its superior performance and wide applicability across various datasets.

**Zero-Shot Temporal Action Detection.** ZSTAD extends TAD to open-set scenarios. To employ ViL models for ZSTAD, the previous methods [8, 18, 19, 20, 28] follow a foreground-based approach and adopt the text prompt tuning strategies. EffPrompt [8] generates action foreground proposals via an off-the-shelf proposal detector (e.g., BMN [13]) and classifies actions utilizing CLIP [21]. STALE [18] and ZEETAD [20] generate the foreground candidate proposals utilizing a class-agnostic representation masking method and parallels localization and classification. Despite the advantage of not relying on an external proposal detector, both methods provide the visual features to the localization branch without fusing with textual features, and their masked foreground representation does not contain text information. Recently introduced UnLoc [26] fuses the video and text information without any pre-extracted proposals. However, the fusion stage of UnLoc proceeds with simple self-attention to align both text and video modalities. In contrast, our approach emphasizes the importance of cross-modal fusion in integrating text and video information across the entire process.

## 3 Method

### 3.1 Problem Definition

Given a training set of untrimmed videos $D_{train}$, each video contains the set of videos features $X = \{x_t\}_{t=1}^{T}$, where $T$ denotes snippets (a few sequences of frames). The labels of each video $Y = \{s_n, e_n, c_n\}_{n=1}^{N}$ contain $N$ action instances, where $s_n$ and $e_n$ represent the start and end points of each instance, and $c_n$ identifies the category of the action. For the zero-shot scenarios, the action categories in $D_{train}$ and $D_{test}$ are distinct and non-overlapping, with $D_{train} \cap D_{test} = \varnothing$. ZSTAD focuses on classifying and localizing actions in untrimmed videos, specifically targeting unseen classes that were not present during the training phase.

## 3.2 Cross-Modal Adaptation Baseline

Utilizing the text-video information in the overall process is crucial in the multi-modal adaptation to transfer the comprehensive semantic representation to zero-shot inference [16]. Inspired by the previous work [11] that emphasizes the necessity of enhancing text-visual fusion, we introduce a cross-modal baseline that updates both text and video features in the fusion step. Specifically, following the previous method [26], we adopt ActionFormer [27] as an action detector.

Given an untrimmed video and text, we extract video features $F_{vid}^{(0)} = \phi_{vid}(X) \in \mathbb{R}^{T \times D}$ with a video encoder $\phi_{vid}(\cdot)$ (e.g., I3D [2], CLIP [21], and R(2+1)D [24]), and text features $F_{txt}^{(0)} \in \mathbb{R}^{C \times D}$ with a text encoder, such as CLIP [21], where $D$ denotes the dimension of the embeddings and $C$ represents the number of classes. We utilize "classname" as the text input without any prefix or contextual text prompt. Fig. 2 (a) shows the structure of our cross-modal decoder, which aggregates both text and video features by learning their joint representation. As shown in Fig. 2 (a), both the initial video features $F_{vid}^{(0)}$ and text features $F_{txt}^{(0)}$ are processed through a series of $L$ layers. Each layer consists of a multi-head self-attention (MHSA), followed by a multi-head cross-attention (X-MHA), and then refined by a feed-forward network (FFN). In MHSA for video features $F_{vid}^{(0)}$, we employ multi-scale visual features with multi-head local self-attention [27]. In each layer $l = 1, \ldots, L$, the features are updated through MHSA, transforming $F_{vid}^{(l-1)}$ and $F_{txt}^{(l-1)}$ to $F'^{(l)}_{vid}$ and $F'^{(l)}_{txt}$, respectively. Then, these updated features are passed through two X-MHA. For simplicity, we denote only the case with video features as the query, but both cases (video features as the query and text features as the query) are utilized. When the video features are utilized as the query and the text features as the key and value, the operation is formulated as follows:

$$F''^{(l)}_{vid} = \text{X-MHA}(F'^{(l)}_{vid}, F'^{(l)}_{txt}) = \text{Softmax}\left(\frac{\text{Q}(F'^{(l)}_{vid})\text{K}(F'^{(l)}_{txt})^{\top}}{\sqrt{d}}\right)\text{V}(F'^{(l)}_{txt}), \quad (1)$$

where $\text{Q}(\cdot)$, $\text{K}(\cdot)$, and $\text{V}(\cdot)$ represent linear projections, and $d$ is the number of head dimension in X-MHA. The output features from X-MHA are denoted as $F''^{(l)}_{vid}$ and $F''^{(l)}_{txt}$. Finally, these updated features via cross-attention are refined by an FFN, resulting in the final updated features for the corresponding layer, denoted as $F_{vid}^{(l)}$ and $F_{txt}^{(l)}$. After processing through all $L$ layers, the final features are denoted as $F_{vid}^{(L)}$ and $F_{txt}^{(L)}$.

To demonstrate the effectiveness of our cross-modal baseline, we compare it with existing foreground-based methods [18, 20] on THUMOS14, as shown in Fig. 2 (b). The detailed description of the experimental setup is described in Sec. 4.1. In the 50%-50% setting, our baseline achieves superior performance compared to STALE [18], which adapts text prompt tuning (TPT), and ZEETAD [20], which applies both text prompt tuning and an Adapter [4]. These results demonstrate that our baseline achieves higher performance without relying on text prompt tuning methods. In contrast, our baseline exhibits sub-optimal performance at the 75%-25% setting compared to ZEETAD [20] with Adapter [4]. As described in Sec. 1, our baseline model struggles with the *common-action bias issue* for actions with shared common sub-actions in the untrimmed videos.

## 3.3 Method

In this section, we introduce the overall process of Text-infused attention and Foreground-aware Action Detection (Ti-FAD). We first describe Text-infused Cross Attention (TiCA) that enables the text features focus on the relevant discriminative visual features. Subsequently, we describe the foreground-aware head that suppresses noisy background information in the untrimmed video. Fig. 3 shows the overview of our proposed framework.

**Text-infused Cross Attention (TiCA).** Our TiCA mainly aims to encourage the model to focus on the discriminative sub-actions by following steps: (1) producing the class score map using video and text features to represent text-related visual parts, (2) selecting top-k points to extract discriminative points, (3) redefining the indices by merging adjacent indices based on a dynamic threshold, (4) generating Gaussian kernels based on the redefined indices to produce Salient Attentive Mask (SAM), and (5) applying the generated SAM to cross-attention.

First, we utilize a temporal class score map that incorporates information from both the text and video by computing the similarity between the video features $F'^{(l)}_{vid}$ and text features $F'^{(l)}_{txt}$ after MHSA.

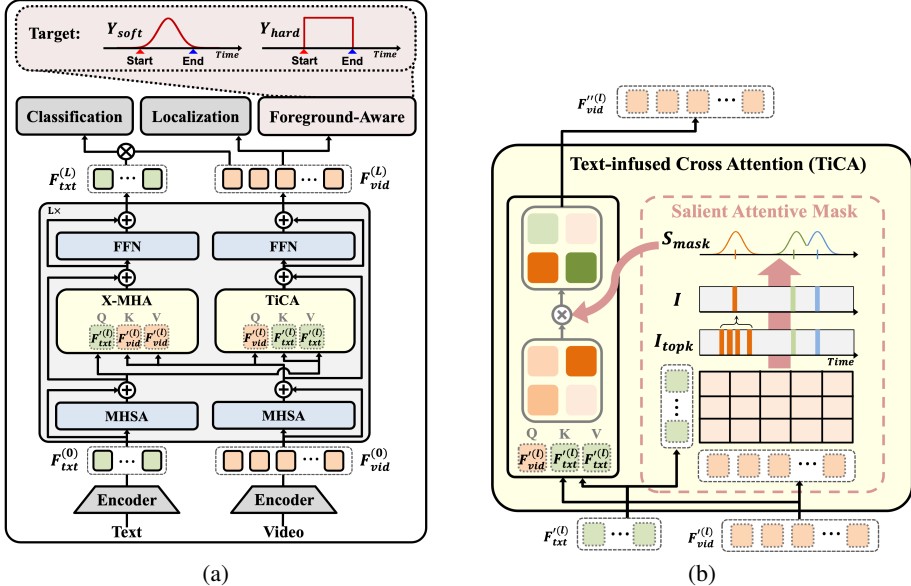

Figure 3: (a) Overview of our proposed Ti-FAD, which includes a foreground-aware head that suppresses the irrelevant background frames, leading the model to concentrate more on the foreground segments, and (b) TiCA that guides the text to focus on the discriminative visual features by employing SAM.

Then, we determine the class-agnostic probability sequence by obtaining the maximum value across the class dimension $C$:

$$P_s = \max_C (F'^{(l)}_{vid} F'^{(l)^\top}_{txt}). \tag{2}$$

The class-agnostic probability sequence $P_s = \{p_t\}_{t=1}^T$ provides a score for each time index, indicating the probability of action occurrence. To produce Gaussian kernels centered on where the action is probably to occur, we define $I_{topk}$ as center location candidates corresponding to the top-$k$ time indices in $P_s$ as follows:

$$I_{topk} = \{i \mid i \in \text{indices of top-}K \text{ in } P_s\}, \tag{3}$$

where $K$ is dynamically determined based on the length of the sequence $T$, calculated as a ratio $r$ of the sequence length with $K = T \times r$.

We redefine the indices set of center location candidates $I_{topk}$ as the average value between close centers if their distance is within a dynamic threshold $\theta(T)$, defined as $\theta(T) = \theta_{base} \cdot T/T_{init}$, where $\theta_{base}$ is the base threshold value, and $T_{init}$ is the initial sequence length. Thus, the redefined discriminative indices set $I$ is calculated as:

$$I = \begin{cases} \frac{i+j}{2} & \text{if } |i-j| < \theta(T) \\ i & \text{otherwise} \end{cases}, \qquad \forall (i,j) \in I_{topk}. \tag{4}$$

This redefined discriminative indices set is utilized to produce SAM value $S_{mask}$ using the kernel function (e.g., Gaussian). To generate a Gaussian kernel, we extract $\sigma \in \{\sigma_i \mid i \in I\}$ from the updated features $F'^{(l)}_{vid}$ with a single linear layer. The sigma $\sigma_i$ values are obtained only from corresponding indices from the redefined discriminative indices set $I$.

To calculate the overall temporal weight across the video sequence, we sum all weights of the Gaussian kernels as: $F(t) = \sum_{i \in I} G(t; i, \sigma_i)$. Subsequently, the SAM value $S_{mask}$ is produced by a normalization process that adjusts $F(t)$ to ensure its maximum value across the temporal dimension is scaled to 1 as follows:

$$S_{mask} = \frac{F(t)}{\max_t (F(t))} \in \mathbb{R}^T. \tag{5}$$

Consequently, we directly integrate the SAM value $S_{mask}$ into the attention score within the multi-head cross-attention X-MHA$(\cdot)$ as defined in Eq. 1, as shown in Fig. 3 (b). This integration dynamically guides the text features to focus on the most relevant visual features. In each layer $l = 1, \ldots, L$,

the X-MHA($\cdot$) is re-defined as Text-infused Cross-Attention (TiCA) and is described as follows:

$$F''^{(l)}_{vid} = \text{TiCA}(F'^{(l)}_{vid}, F'^{(l)}_{txt}) = \text{Softmax}\left(\frac{\text{Q}(F'^{(l)}_{vid})\text{K}(F'^{(l)}_{txt})^\top}{\sqrt{d}} \cdot (S_{mask} \otimes \mathbf{1}_C)\right)\text{V}(F'^{(l)}_{txt}), \quad (6)$$

where $\mathbf{1}_C \in \mathbb{R}^C$ denotes a vector of ones with dimensions corresponding to the class dimension. The term $S_{mask} \otimes \mathbf{1}_C \in \mathbb{R}^{T \times C}$ denotes the element-wise multiplication of SAM with a matrix of ones, broadcast across the class dimension.

**Foreground-Aware Head.** Our foreground-aware head aims to suppress irrelevant background information and enhance the model's focus on foreground action segments. Given the updated video feature $F^{(L)}_{\text{vid}}$, it is fed into a lightweight network with three layers of 1D convolutions after each feature level, inspired by the previous method [27]. Subsequently, the sigmoid activation function is applied to ensure that the values range from 0 to 1. This process produces two types of outputs: $A^{fg}_{soft} \in \mathbb{R}^T$ and $A^{fg}_{hard} \in \mathbb{R}^T$, representing the soft and hard foreground-awareness scores. Depending on the target values, we introduce two types of foreground-aware heads: soft foreground-aware head (S-FAD) and hard foreground-aware head (H-FAD), both designed to predict the actionness score of each snippet in the time location.

For S-FAD, the target actionness score represents the normalized distance between time step $i$ and the center of the corresponding ground-truth action segment. We define the actionness target score using a Gaussian function $\phi(x, \mu, \sigma)$ calculated as follows:

$$Y_{soft} = \min\left(\phi(l; 0, \sigma), \ \phi(r; 0, \sigma)\right), \ \ l = |c - s| \text{ and } r = |c - e|, \quad (7)$$

where $l$ and $r$ represent the distances from time step $i$ to the start $s$ and end $e$ points of the action, respectively. The mean $\mu$ is set to 0, centering the distribution around the action center, while $\sigma$ is a standard deviation that is a hyperparameter to control the spread of the Gaussian distribution.

For H-FAD, we assign 1 for the foreground and 0 for the background segments. This binary setup directly reflects the presence or absence of action within each time snippet, making it straightforward for the model to distinguish action segments from non-action segments. The target value $Y_{hard}$ is defined as follows:

$$Y_{hard} = \mathbb{1}_{[s \leq i \leq e]}. \quad (8)$$

To optimize both aspects simultaneously, we compute the foreground-aware loss $\mathcal{L}_{fg}$ as follows:

$$\mathcal{L}_{fg} = \alpha\mathcal{L}_{soft}(A^{fg}_{soft}, Y_{soft}) + \beta\mathcal{L}_{hard}(A^{fg}_{hard}, Y_{hard}), \quad (9)$$

where both loss functions $\mathcal{L}_{soft}$ and $\mathcal{L}_{hard}$ utilize binary cross-entropy loss. Here, $\alpha$ and $\beta$ are hyper-parameters to control trade-off. During inference, we multiply both scores $A^{fg}_{soft}$ and $A^{fg}_{hard}$ to the classification scores to suppress the irrelevant information of the actions.

### 3.4 Training and Inference

**Training.** For localization, we utilize the localization head similar to the foreground-aware head, but it predicts the distance $(d^s_t, d^e_t)$ from time step $t$ to the action boundaries following the anchor-free method [27]. We optimize the distances to the action boundaries with the DIoU loss [30], denoted as $\mathcal{L}_{loc}$. For classification, we compute similarity scores as followed by the standard zero-shot settings [8, 18, 20, 26, 29]. Specifically, the updated text features $F^{(L)}_{txt}$ are fed into simple projection $\Phi_{proj}(\cdot)$ and the classification output is computed as: $C_{cls} = \Phi_{proj}(F^{(L)}_{txt}) \cdot F^{(L)}_{vid}{}^\top$, where $C_{cls} \in \mathbb{R}^{T \times C}$ feature snippet represents the probability of action categories at time step $t$. We employ the focal loss [15] to recognize action categories, denoted as $\mathcal{L}_{cls}$. The total objective function combined with the foreground-aware loss discussed in Sec. 3.3 is defined as follows: $\mathcal{L} = \sum\left(\frac{\lambda_c}{T_+}\mathcal{L}_{cls} + \frac{\lambda_l}{T_+}\mathcal{L}_{loc} + \frac{\lambda_f}{T_+}\mathcal{L}_{fg}\right)$, where $T_+$ represents the total number of positive samples. The $\lambda_c$, $\lambda_l$, and $\lambda_f$ represent the hyper-parameters to control trade-off.

**Inference.** The model infers $(d^s_t, d^e_t, p(c_t))$ at each time step $t$. The $d^s_t$ and $d^e_t$ are the distance from time step $t$ to the start and end points, respectively. Here, $p(c_t)$ is the action confidence score. The redundant proposals are removed by SoftNMS [1] to obtain the final action.

Table 1: Performance comparison with the state-of-the-art methods on THUMOS14 and ActivityNet v1.3. Average mAP is calculated to evaluate performance at different tIoU thresholds: [0.3:0.1:0.7] for THUMOS14 and [0.5:0.05:0.95] for ActivityNet v1.3. $^\dagger$ represents using the prompt ensemble.

| Train Split | Model | Prompt Tuning | Feature | | THUMOS14 | | | | | | ActivityNet v1.3 | | | |
| --- | --- | --- | --- | --- | --- | --- | --- | --- | --- | --- | --- | --- | --- | --- |
| | | | Video | Text | 0.3 | 0.4 | 0.5 | 0.6 | 0.7 | Avg | 0.5 | 0.75 | 0.95 | Avg |
| 50% Seen 50% Unseen | B-II [18] | ✓ | CLIP-B [21] | CLIP-B [21] | 21.0 | 16.4 | 11.2 | 6.3 | 3.2 | 11.6 | 25.3 | 13.0 | 3.7 | 12.9 |
| | B-I [18] | ✓ | CLIP-B [21] | CLIP-B [21] | 27.2 | 21.3 | 15.3 | 9.7 | 4.8 | 15.7 | 28.0 | 16.4 | 1.2 | 16.0 |
| | EffPrompt [8] | ✓ | I3D [2] | CLIP-B[21] | 37.2 | 29.6 | 21.6 | 14.0 | 7.2 | 21.9 | 32.0 | 19.3 | 2.9 | 19.6 |
| | STALE [18] | ✓ | I3D [2] | CLIP-B [21] | 38.3 | 30.7 | 21.2 | 13.8 | 7.0 | 22.2 | 32.1 | 20.7 | **5.9** | 20.5 |
| | UnLoc-B [26] | ✗ | CLIP-B [21] | CLIP-B [21] | - | - | - | - | - | - | 36.9 | - | - | - |
| | UnLoc-L [26] | ✗ | CLIP-L [21] | CLIP-L [21] | - | - | - | - | - | - | 43.2 | - | - | - |
| | UnLoc-L [26]$^\dagger$ | ✗ | CLIP-L [21] | CLIP-L [21] | - | - | - | - | - | - | 43.7 | - | - | - |
| | ZEETAD [20] | ✓ | I3D [2] | CLIP-B [21] | 45.2 | 38.8 | 30.8 | 22.5 | 13.7 | 30.2 | 39.2 | 25.7 | 3.1 | 24.9 |
| | **Baseline (Ours)** | ✗ | I3D [2] | CLIP-B [21] | 55.9 | 47.5 | 35.9 | 22.7 | 10.5 | 34.5 | 44.1 | 26.1 | 3.1 | 26.6 |
| | **Ti-FAD (Ours)** | ✗ | CLIP-B [21] | CLIP-B [21] | 44.3 | 36.8 | 27.7 | 18.2 | 9.4 | 27.3 | 50.4 | 32.1 | 5.2 | 31.7 |
| | **Ti-FAD (Ours)** | ✗ | CLIP-L [21] | CLIP-L [21] | 43.7 | 36.3 | 27.6 | 18.2 | 10.0 | 27.2 | 51.0 | 32.6 | 4.8 | 32.1 |
| | **Ti-FAD (Ours)** | ✗ | I3D [2] | CLIP-B [21] | **57.0** | **51.4** | **43.3** | **33.0** | **21.2** | **41.2** | 50.6 | 32.2 | 5.2 | 32.0 |
| | **Ti-FAD (Ours)** | ✗ | I3D [2] | CLIP-L [21] | 56.7 | 50.8 | 42.4 | 32.3 | 20.8 | 40.6 | **50.8** | **32.6** | **5.6** | **32.3** |
| 75% Seen 25% Unseen | B-II [18] | ✓ | CLIP-B [21] | CLIP-B [21] | 28.5 | 20.3 | 17.1 | 10.5 | 6.9 | 16.6 | 32.6 | 18.5 | 5.8 | 19.6 |
| | B-I [18] | ✓ | CLIP-B [21] | CLIP-B [21] | 33.0 | 25.5 | 18.3 | 11.6 | 5.7 | 18.8 | 35.6 | 20.4 | 2.1 | 20.2 |
| | EffPrompt [8] | ✓ | I3D [2] | CLIP-B [21] | 39.7 | 31.6 | 23.0 | 14.9 | 7.5 | 23.3 | 37.6 | 22.9 | 3.8 | 23.1 |
| | STALE [18] | ✓ | I3D [2] | CLIP-B [21] | 40.5 | 32.3 | 23.5 | 15.3 | 7.6 | 23.8 | 38.2 | 25.2 | 6.0 | 24.9 |
| | UnLoc-B [26] | ✗ | CLIP-B [21] | CLIP-B [21] | - | - | - | - | - | - | 40.2 | - | - | - |
| | UnLoc-L [26] | ✗ | CLIP-L [21] | CLIP-L [21] | - | - | - | - | - | - | 47.4 | - | - | - |
| | UnLoc-L [26]$^\dagger$ | ✗ | CLIP-L [21] | CLIP-L [21] | - | - | - | - | - | - | 48.8 | - | - | - |
| | ZEETAD [20] | ✓ | I3D [2] | CLIP-B [21] | 61.4 | 53.9 | 44.7 | 34.5 | 20.5 | 43.2 | 51.0 | 33.4 | 5.9 | 32.5 |
| | **Baseline (Ours)** | ✗ | I3D [2] | CLIP-B [21] | 59.1 | 50.9 | 39.3 | 25.7 | 12.2 | 37.4 | 47.0 | 27.2 | 3.8 | 28.3 |
| | **Ti-FAD (Ours)** | ✗ | CLIP-B [21] | CLIP-B [21] | 47.6 | 40.0 | 30.5 | 19.8 | 10.5 | 29.7 | 53.6 | 34.1 | 5.4 | 33.8 |
| | **Ti-FAD (Ours)** | ✗ | CLIP-L [21] | CLIP-L [21] | 48.3 | 40.9 | 30.8 | 21.1 | 11.7 | 30.6 | 54.5 | 35.2 | 5.3 | 34.5 |
| | **Ti-FAD (Ours)** | ✗ | I3D [2] | CLIP-B [21] | 64.0 | 58.5 | **49.7** | 37.7 | 24.1 | 46.8 | **53.8** | 34.8 | **7.0** | **34.7** |
| | **Ti-FAD (Ours)** | ✗ | I3D [2] | CLIP-L [21] | **64.8** | **58.8** | 49.6 | **38.3** | **24.3** | **47.3** | 53.7 | **34.9** | 6.0 | 34.4 |

# 4 Experiments

## 4.1 Experiment Settings

**Datasets.** We conduct experiments on THUMOS14 [7], and ActivityNet v1.3 [5], two benchmark datasets commonly used in ZSTAD. THUMOS14 consists of 20 sports action classes, containing 200 training and 213 testing videos. ActivityNet v1.3 contains 200 daily action classes and a total of 19,994 videos. We split the training, validation, and testing in the ratio 2:1:1 according to the standard setting [8]. For the zero-shot setting, we assume two situations as in [8]: 50%-50% setting (training for 50% of the action categories and testing for the remaining 50%), 75%-25% setting (training for 75% of the action categories and testing for the remaining 25%). To make it statistically robust, we utilize 10 random splits and average them in the final performance.

**Evaluation Metric.** We utilize mean Average Precision (mAP) as an evaluation metric. The mAP is calculated by averaging the precisions at different temporal intersection over union (tIoU) thresholds. The tIoU thresholds are set at intervals of 0.1 from 0.3 to 0.7 ([0.3:0.1:0.7]) for THUMOS14, and at intervals of 0.05 from 0.5 to 0.95 ([0.5:0.05:0.95]) for ActivityNet v1.3.

**Implementation Details.** Following the existing methods [8, 18, 20, 26], we employ two-stream I3D [2] and CLIP [21] models to extract video features. The two-stream I3D features are concatenated for the video inputs. For THUMOS14, video features are extracted from 16 frames as a segment using a sliding window with a stride of 4. For ActivityNet v1.3, features are extracted with a stride of 16 and downsampled to 128 dimensions. For the text encoder, we adopt the frozen pre-trained CLIP model [21] as the text encoder, using ViT-B/16 and ViT-L/14. Our model is trained for 20 epochs on THUMOS14 and 15 epochs on ActivityNet v1.3 using Adam with 5 epochs of linear warmup. The initial learning rate is 0.0001, updated with MultiStepLR for THUMOS14 and cosine annealing [17] for ActivityNet v1.3, respectively. The hyperparameters for $\alpha$, $\beta$, $\lambda_l$, and $\lambda_f$ are set to 1, while $\lambda_c$ is set to 0.5 for all datasets. All experiments are conducted with a single NVIDIA RTX A6000 GPU.

## 4.2 Main Results

We compare our cross-modal baseline and Ti-FAD with state-of-the-art ZSTAD methods [8, 18, 20, 26] by computing the average mAP at different tIoUs. Table 1 shows the performance comparison on

Table 2: Analysis of contributions of each component on THUMOS14.

| Row | Method | SAM | S-FAD | H-FAD | mAP@AVG | |
|---|---|---|---|---|---|---|
| | | | | | 50%-50% | 75%-25% |
| 1 | Baseline | | | | 34.5 | 37.4 |
| 2 | | ✓ | | | 35.3 | 40.0 |
| 3 | | | ✓ | | 36.1 | 39.9 |
| 4 | | | | ✓ | 38.4 | 43.0 |
| 5 | Ti-FAD | ✓ | ✓ | | 35.9 | 40.1 |
| 6 | | | ✓ | ✓ | 37.4 | 43.2 |
| 7 | | ✓ | | ✓ | 40.2 | 45.2 |
| 8 | | ✓ | ✓ | ✓ | **41.2** | **46.8** |

Table 3: Analysis of X-MHA operation in TiCA module on THUMOS14.

| Method | mAP@tIoU (%) | | | | | | | |
|---|---|---|---|---|---|---|---|---|
| | 50%-50% | | | | 75%-25% | | | |
| | 0.3 | 0.5 | 0.7 | Avg | 0.3 | 0.5 | 0.7 | Avg |
| Baseline | 55.9 | 35.9 | 10.5 | 34.5 | 59.1 | 39.3 | 12.2 | 37.4 |
| Text-to-vision | **57.4** | 42.5 | 20.0 | 40.6 | 62.3 | 47.8 | 23.6 | 45.4 |
| Vision-to-text (Ours) | 57.0 | **43.3** | **21.2** | **41.2** | **64.0** | **49.7** | 24.1 | **46.8** |
| Both | 56.2 | 42.8 | **21.2** | 40.7 | 63.6 | 48.8 | **24.5** | 46.5 |

THUMOS14 and ActivityNet v1.3. Our baseline achieves superior performance for the 50%-50% setting, indicating the robustness of our baseline, as it effectively integrates text information, enabling better adaptation to unseen scenarios. However, for the 75%-25% setting, the performance of our baseline is marginally inferior to ZEETAD [20]. In contrast, Ti-FAD with I3D features achieves state-of-the-art performance for both the 50%-50% and 75%-25% settings, particularly at tIoU thresholds of 0.5, 0.6, and 0.7 on THUMOS14. Even with CLIP as a video encoder, Ti-FAD still shows competitive performance compared to the previous methods using I3D features.

### 4.3 Further Analysis

**Component Analysis of Ti-FAD.** Table 2 shows a comprehensive component analysis of our Ti-FAD on THUMOS14. The table evaluates the impact of different components, including SAM, S-FAD, and H-FAD on the performance of the model. The baseline performance without any additional components is reported in Row 1. Rows 2-4 evaluate the contribution of each individual component. As shown in Table 2, we first observe that employing SAM (Row 2) to the cross-modal alignment process of the baseline (Row 1) enhances the performance by +0.8% for the 50%-50% setting and +2.6% for the 75%-25% setting, which means that applying only SAM contributes to better performance. In Row 3 and 4, we recognize that S-FAD increases the baseline performance by 1.6%, and H-FAD by 3.9% for the 50%-50% setting. These results indicate that S-FAD and H-FAD encourage a background suppression effect by focusing on the class-agnostic action features in the untrimmed videos. Particularly, in the 75%-25% setting, incorporating a H-FAD that emphasizes the foreground to SAM enhances the performance of the baseline from 37.4% to 45.2% (Row 7). It indicates the need to separate the foreground from the background. When incorporating all the components, our model boosts the best performance at 41.2% and 46.8% in 50%-50% and in 75%-25% settings, respectively.

**Analysis of X-MHA Operation in TiCA Module.** We investigate the impact of the X-MHA direction in the TiCA module, as shown in Table 3. We compare the performance with three approaches: using $S_{mask}$ as an attention bias for text-to-vision cross-attention, vision-to-text cross-attention, and both directions of cross-attention. To apply text-to-vision cross-attention, we transpose $S_{mask} \otimes \mathbf{1}_C$ in Eq. (6). Our model shows similar performance regardless of whether $S_{mask}$ is used for text-to-vision or vision-to-text cross-attention. The results indicate that the most crucial part of our TiCA is SAM, and the choice between text-to-vision and vision-to-text cross-attention is less critical.

Table 4: Ablation studies of our Ti-FAD on THUMOS14 in the 50%-50% setting.

(a) Kernel design of SAM.

| Kernel | mAP@tIoU (%) | | | |
|---|---|---|---|---|
| | 0.3 | 0.5 | 0.7 | Avg |
| Tophat | 57.0 | 43.0 | 20.5 | 40.9 |
| Cauchy | **57.7** | 43.0 | 20.6 | 41.1 |
| **Gaussian** | 57.0 | **43.3** | **21.2** | **41.2** |

(b) Structure of SAM.

| Type | mAP@tIoU(%) | | | |
|---|---|---|---|---|
| | 0.3 | 0.5 | 0.7 | Avg |
| Sum | 57.1 | 42.1 | 20.1 | 40.5 |
| Mul | 56.7 | 42.9 | 20.8 | 40.9 |
| Wsum | **58.0** | 42.9 | 20.4 | 41.0 |
| **Wmul** | 57.0 | **43.3** | **21.2** | **41.2** |

(c) Sigma ($\sigma$) of Gaussian mask.

| Sigma ($\sigma$) | mAP@tIoU (%) | | | |
|---|---|---|---|---|
| | 0.3 | 0.5 | 0.7 | Avg |
| $\sigma = 0.1$ | 56.6 | 43.1 | 21.1 | 40.9 |
| $\sigma = 0.5$ | 56.8 | 42.9 | 20.9 | 40.8 |
| $\sigma = 1.0$ | 56.8 | 43.1 | **21.2** | 41.1 |
| **Learnable** | 57.0 | 43.3 | 21.2 | 41.2 |

(d) Text prompt tuning (TPT).

| Type | mAP@tIoU (%) | | | |
|---|---|---|---|---|
| | 0.3 | 0.5 | 0.7 | Avg |
| EffPrompt [8] | 56.7 | 42.1 | 20.3 | 40.4 |
| TPT (50) [10] | 56.6 | 42.9 | 20.2 | 40.5 |
| TPT (100) [10] | **57.1** | **43.3** | 21.0 | 41.1 |
| **w/o TPT** | 57.0 | **43.3** | **21.2** | **41.2** |

**Ablations on Kernel Design of SAM.** In Table 4 (a), we experiment with different kernel designs for SAM to understand their impact on performance. We test the kernels including Tophat, Cauchy, and Gaussian. The results indicate that our approach is robust to the design of the kernel since Ti-FAD achieves similar performance utilizing each kernel. The Gaussian kernel consistently yields the best result with an average mAP of 41.2%, thereby we select the Gaussian kernel for our final model.

**Ablations on the Structure of SAM.** We investigate the impact of different application structures for combining SAM with visual features in Table 4 (b). Sum and Mul indicate the structure of reflecting the mask on the visual feature fed as input to the X-MHA operation, via element-wise sum (Row 1) and element-wise multiplication (Row 2). Wsum and Wmul represent the application of a mask to attention weights within the X-MHA operation by adding (Row 3) and multiplying (Row 4). We observe that the performance of Ti-FAD is not substantially influenced by different types of masks, with the best performance achieved by multiplying the mask by the attention weights.

**Ablations on Sigma of Gaussian.** We analyze the performance of different $\sigma$ values, which determine the width of the Gaussian kernel. As shown in Table 4 (c), we experiment with fixed $\sigma$ values of 0.1, 0.5, and 1.0, as well as a learnable $\sigma$. Our findings indicate that allowing $\sigma$ to be a learnable parameter yields the best performance, with an average mAP of 41.2%. A learnable $\sigma$ allows the model to dynamically adjust to the specific temporal characteristics of each action. During our experiments, we observe that the learnable $\sigma$ values converge within the range of 0.1 to 1.

**Ablations on the Strategy of Text Prompt Tuning.** We investigate the effect of different text prompt tuning (TPT) strategies on the performance of Ti-FAD in Table 4 (d). We compare the performance of Ti-FAD with EffPrompt [8], TPT (50) [10], and TPT (100) [10] methods that add learnable embeddings of size 50 and 100, respectively, after the text embeddings on CLIP text encoder. Additionally, we evaluate the model without any TPT. The results demonstrate that Ti-FAD maintains increased performance regardless of the text prompt tuning strategy. Additionally, our model achieves the highest average mAP of 41.2% without text prompt tuning. This observation highlights the ability of Ti-FAD to effectively leverage textual information without the need for extensive prompt tuning.

**Per-Unseen Class AP Comparison.** Fig. 4 demonstrates the per-class Average Precision (AP) comparison between our baseline model and Ti-FAD at a tIoU threshold of 0.5 on THUMOS14. Ti-FAD outperforms the baseline, particularly in action categories that include common sub-actions (e.g., *Running*). For example, in the *PoleVault* category, Ti-FAD achieves an AP of 77.23%, significantly higher than the baseline's 59.63%. Similarly, in the *LongJump* category, Ti-FAD's AP is 79.58%, compared to the baseline's 48.12%. These improvements highlight the ability of Ti-FAD to focus on discriminative sub-actions and suppress irrelevant background information.

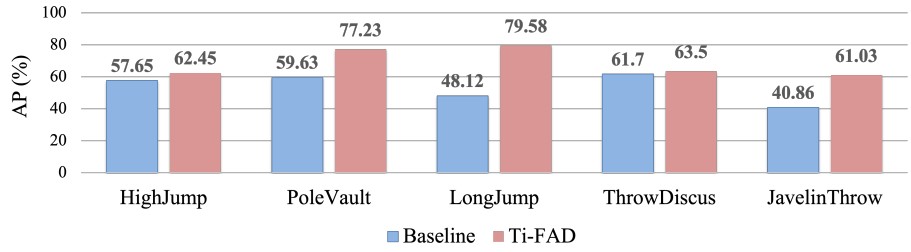

Figure 4: Per-unseen class AP (%) at tIoU threshold 0.5 on THUMOS14.

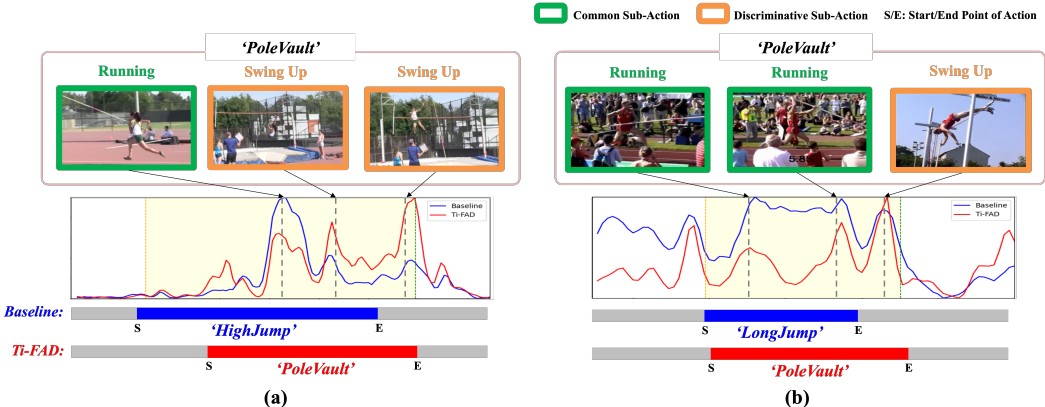

Figure 5: Visualization of the detection results on THUMOS14 in the 50%-50% setting.

## 4.4 Qualitative Results

Fig. 5 shows visual examples of an unseen class *PoleVault* with common sub-actions (e.g., *Running*). In both Fig. 5 ((a)-(b)), our baseline (blue line) over-focuses on the common sub-action (e.g., *Running*), leading to incorrect detection: *HighJump* in (a) and *LongJump* in (b). In contrast, our Ti-FAD (red line) accurately detects the action based on the visual feature of the discriminative sub-action (e.g., *Swing Up*). By capturing the discriminative sub-action associated with the text information, Ti-FAD effectively identifies the unseen class. This ability to integrate text and visual features throughout the detection process significantly improves the model's performance in distinguishing between similar actions that share common sub-actions.

## 5 Conclusion

In this paper, we propose Text-infused and Foreground-aware Action Detection (Ti-FAD), a novel approach designed to address a common-action bias issue in ZSTAD by enhancing the integration of text and visual information. Ti-FAD achieves this through Text-infused Cross Attention (TiCA), which strengthens the model's ability to capture text-related sub-actions and a foreground-aware head that differentiates action segments from the background, leading to an accurate detection process. Our extensive experiments show that Ti-FAD effectively addresses a common-action bias issue and achieves superior performance compared to state-of-the-art ZSTAD methods on THUMOS14 and ActivityNet v1.3. Despite the significant improvements, Ti-FAD's scalability to larger datasets and real-time processing capabilities remains to be fully explored. Additionally, extending our approach to incorporate other modalities, such as audio or depth information, can be studied to enhance performance.

**Acknowledgement.** This work was supported by Institute of Information & communications Technology Planning & Evaluation (IITP) grant funded by the Korea government(MSIT) (No. RS-2019-II190079, Artificial Intelligence Graduate School Program (Korea University), No. 2021-0-02068, Artificial Intelligence Innovation Hub, No. RS-2024-00457882, AI Research Hub Project, and No. RS-2024-00336673, AI Technology for Interactive Communication of Language Impaired Individuals).

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

# Text-Infused Attention and Foreground-Aware Modeling for Zero-Shot Temporal Action Detection - Appendix

Table A: Ablation of cross-modal baseline on THUMOS14.

| | Type | mAP@tIoU(%) | | | |
|---|---|---|---|---|---|
| | | 0.3 | 0.5 | 0.7 | Avg |
| (a) | ActionFormer W/o Cross-Modal Fusion | 30.3 | 15.1 | 3.5 | 16.1 |
| (b) | Cross-Modal Baseline (Self-Attn) | 42.2 | 24.0 | 5.5 | 23.8 |
| (c) | **Cross-Modal Baseline (Cross-Attn)** | **55.9** | **35.9** | **10.5** | **34.5** |

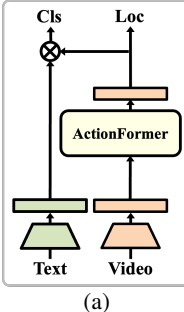 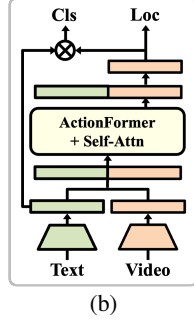 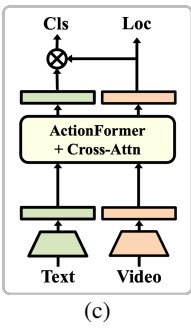

Figure A: Illustrations of different baseline architectures: (a) ActionFormer [27] w/o cross-modal fusion, (b) Cross-modal baseline (self-attn), and (c) (Our baseline) Cross-modal baseline (cross-attn)

## A    Additional Experiments

**Ablation of Cross-Modal Baseline.** To provide a fairer view of our cross-modal part, we conduct a comparative analysis of different baseline architectures in Table A. We compare ActionFormer [27] without the cross-modal fusion and our cross-modal baselines on THUMOS14 in the 50%-50% setting. This table shows that ActionFormer [27] without cross-modal fusion (a) exhibits inferior performance because text and video are not properly aligned, highlighting the importance of cross-modal fusion. Furthermore, the cross-attention baseline (c) outperforms the self-attention baseline (b). To facilitate a clearer understanding of our experimental models, we additionally provide illustrations of (a), (b), and (c) in Fig. A.

Table B: Analysis of X-MHA operation in TiCA module on ActivityNet v1.3.

| Method | mAP@tIoU (%) | | | | | | | |
|---|---|---|---|---|---|---|---|---|
| | 50%-50% | | | | 75%-25% | | | |
| | 0.5 | 0.75 | 0.95 | Avg | 0.5 | 0.75 | 0.95 | Avg |
| Baseline | 44.1 | 26.1 | 3.1 | 26.6 | 47.0 | 27.2 | 3.8 | 28.3 |
| Text-to-vision | 50.8 | 31.9 | **5.6** | 32.0 | **53.9** | 34.5 | 6.3 | 34.3 |
| Vision-to-text (Ours) | 50.6 | **32.2** | 5.2 | 32.0 | 53.8 | **34.8** | **7.0** | **34.7** |
| Both | **50.9** | 32.1 | **5.6** | **32.1** | **53.9** | 34.4 | 6.1 | 34.2 |

**Analysis of X-MHA Operation in TiCA Module.** We investigate the impact of the X-MHA direction in the TiCA module on ActivityNet v1.3, as shown in Table B. We compare the performance with three approaches under the same experimental settings used for THUMOS14. For ActivityNet v1.3, our model also shows similar performance regardless of whether $S_{mask}$ is used for text-to-vision or vision-to-text cross-attention.

Table C: Comparison with other methods for text prompt engineering on THUMOS14.

| Text Prompt | | Basline | | | | Ti-FAD | | | |
|---|---|---|---|---|---|---|---|---|---|
| | | 0.3 | 0.5 | 0.7 | Avg | 0.3 | 0.5 | 0.7 | Avg |
| (a) | "a video of action {classname}" | 53.8 | 33.9 | 9.7 | 32.8 | 55.6 | 42.6 | 20.9 | 40.4 |
| (b) | Prompt Augmentation [21] | 54.3 | 34.6 | 10.3 | 33.4 | 56.3 | 42.7 | 20.6 | 40.5 |
| (c) | Prompt Ensemble [21] | 53.8 | 34.3 | 10.2 | 33.1 | 56.8 | 43.0 | 20.3 | 40.7 |
| (d) | **"{classname}"** | **55.9** | **35.9** | **10.5** | **34.5** | **57.0** | **43.3** | **21.2** | **41.2** |

**Comparison with Other Methods for Text Prompt Engineering.** We compare the performance of our baseline and Ti-FAD with the most utilized types of text prompts following previous works [18, 8, 26, 20] on THUMOS14 in Table C. (a) is used in [18, 8, 20]. (b) refers to using the 28 templates of the prompt, used in UnLoc [26]. (c) refers to using the average embedding vector from the 28 templates at the inference, used in UnLoc [26]. The result demonstrates that Ti-FAD shows similar performance regardless of the type of text prompt. These results show that Ti-FAD's performance does not depend on the type of text prompt.

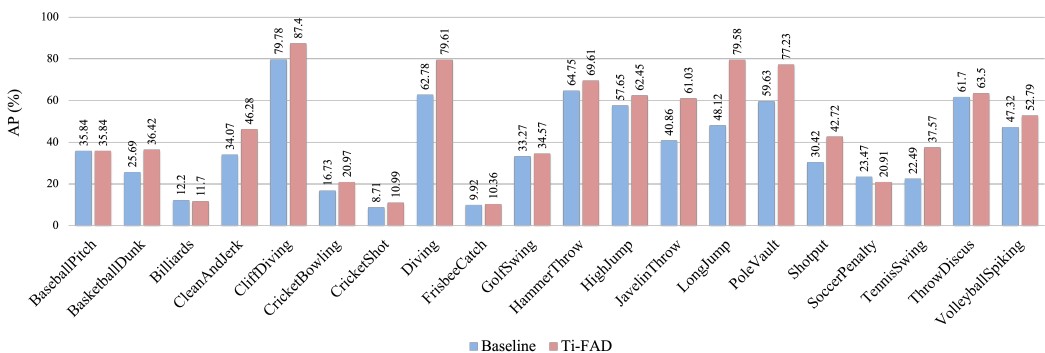

Figure B: Per-unseen class AP (%) at tIoU threshold 0.5 on THUMOS14.

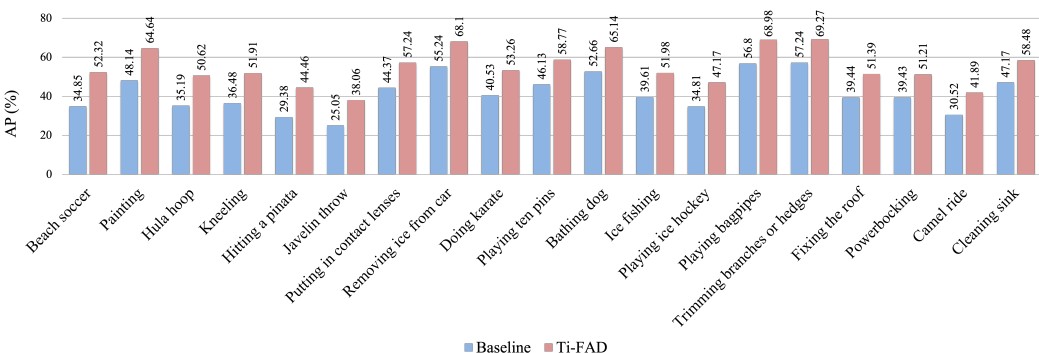

Figure C: Per-unseen class AP (%) at tIoU threshold 0.5 on ActivityNet v1.3.

**Additional Per-Unseen Class AP Comparison.** Fig. B shows the per-unseen class Average Precision (AP) at an tIoU threshold of 0.5 on THUMOS14. The comparison between the baseline and Ti-FAD models illustrates the effectiveness of our approach in improving detection accuracy across various unseen classes. Ti-FAD demonstrates substantial performance improvements over the baseline model, particularly in action categories that involve common sub-actions (e.g., *Running*). This indicates Ti-FAD's enhanced capability to capture and focus on discriminative sub-actions while effectively suppressing irrelevant background information. For instance, in the *PoleVault* category, Ti-FAD achieves an AP of 77.23%, significantly higher than the baseline's 59.63%. Similarly, in the *LongJump* category, Ti-FAD's AP is 79.58%, compared to the baseline's 48.12%. In the *Diving* category, Ti-FAD shows an impressive AP of 79.61%, compared to the baseline's 62.78%. For *Shotput*, Ti-FAD achieves an AP of 42.72%, outperforming the baseline's 30.42%. These results illustrate Ti-FAD's robustness and effectiveness in dealing with unseen action categories, showcasing its ability to generalize better in zero-shot scenarios. Furthermore, we provide additional per-unseen class AP results for 20 classes of ActivityNet v1.3 compared with our cross-modal baseline in Fig. C. We observe that sharing common sub-actions

between multiple classes also occurs on ActivityNet v1.3. For example, actions such as *Painting*, *Cleaning sink*, *Washing face*, *Cleaning windows*, *Hand car wash*, *Cleaning shoes*, *Ironing clothes*, *Hand washing clothes*, and *Washing dishes* share the common sub-action part *Wiping motion with hands*, leading to similar issue. These results demonstrate that the common sub-action bias issue is not isolated to THUMOS14.

