# OpenReview forum: "Text-Infused Attention and Foreground-Aware Modeling for Zero-Shot Temporal Action Detection"
_NeurIPS.cc/2024/Conference — NeurIPS 2024 poster_

### Official Review · Reviewer_mst5 · 2024-07-12

**Soundness:** 4
**Presentation:** 4
**Contribution:** 4
**Rating:** 6
**Confidence:** 3

**Summary:**

This paper proposes a Ti-FAD framework for the zero-shot temporal action detection (ZSTAD) task, where the goal is to locate and classify unknown action classes. The proposed Ti-FAD features a mutual corss=attention integration module for detection, and leverages text-related sub-action information to mitigate the action bias problem in ZSTAD. Strong performances are achieved on THUMOS14 and ActivityNet v1.3.

**Strengths:**

1. The motivation is strong and persuasive. However, the common sub-action bias may mislead the classification. This problem greatly affects the performance of visual-only and cross-modal methods.
2. The proposed text-infused cross attention and foreground-aware head are technically sound, and demonstrate strong effectiveness in the ablation study.
3. The paper is overall well-written and easy-to-read.
4. The final performances are strong, achieving state-of-the-art performances on THUMOS14 and ActivityNet v1.3.

**Weaknesses:**

This paper is overall of good quality. I do not spot major technical weaknesses.

1. What exactly is the text used for the text feature extraction? The authors only mention they use a text encoder for text feature extraction (lines 105-106) but do not mention the textual prompts.
2. Baseline performance. The authors propose a strong cross-modal baseline in Sec 3.2, whose performance already outperforms several existing methods (Fig. 2 (b)). Can the authors provide a more detailed breakdown to identify the most effective part of the baseline?
3. The authors only use I3D visual feature for experiments in Table 1, it would be help to evaluate with CLIP features to demonstrate the generalization ability to different features.
4. Limitations are not discussed in the paper.

**Questions:**

1. What are the textual prompts used for text feature extraction?
2. What are the most important designs in the baseline to make it outperform existing methods?

**Limitations:**

Limitations are not discussed.

---

> ### Author Rebuttal · Authors · 2024-08-06
>
> > **W1,Q1) What exactly is the text used for the text feature extraction? The authors only mention they use a text encoder for text feature extraction (Lines 105-106) but do not mention the textual prompts.**
>
> We simply use "{class name}" as the text prompt without any prefix or contextual text.
> We will add a detailed description of the text prompt in Line 107 to provide a clear view.
> Furthermore, to show that Ti-FAD's performance does not depend on the type of text prompt, we compare the performance of our baseline and Ti-FAD with the most utilized types of text prompts following previous works [2,3,4,6] on THUMOS14 in the 50%-50% setting.
>
> ||Text Prompt|Baseline mAP@0.3|mAP@0.5|mAP@0.7|mAP@Avg|Ti-FAD mAP@0.3|mAP@0.5|mAP@0.7|mAP@Avg|
> |-|-|-|-|-|-|-|-|-|-|
> |(a)|"a video of action {classname}"|53.8|33.9|9.7|32.8|55.6|42.6|20.9|40.4|
> |(b)|Prompt Augmentation [5]|54.3|34.6|10.3|33.4|56.3|42.7|20.6|40.5|
> |(c)|Prompt Ensemble [5]|53.8|34.3|10.2|33.1|56.8|43.0|20.3|40.7|
> |(d)|**"{classname}"**|**55.9**|**35.9**|**10.5**|**34.5**|**57.0**|**43.3**|**21.2**|**41.2**|
>
> (a) is utilized in [2,3,6].
> (b) refers to using the 28 templates of the prompt, utilized in UnLoc [4].
> (c) refers to using the average embedding vector from the 28 templates at the inference, utilized in UnLoc [4].
> The result demonstrates that our Ti-FAD shows similar performance regardless of the type of text prompt.
> We will add this experiment to the Appendix.
>
> > **W2,Q2) Baseline performance. The authors propose a strong cross-modal baseline in Sec 3.2, whose performance already outperforms several existing methods (Fig. 2 (b)). Can the authors provide a more detailed breakdown to identify the most effective part of the baseline?**
>
> The most effective part of our baseline is integrating text and visual information throughout the entire detection process.
> In Lines 31-33, most existing methods [2,3,6] utilize pre-extracted visual features, which are not integrated with textual information throughout the entire detection process.
> This separated process limits their ability to fully leverage the contextual information between both modalities.
>
> To demonstrate the effectiveness of cross-modal fusion in our baseline, we conduct a comparative analysis of ActionFormer without the cross-modal fusion and our cross-modal baselines.
> To facilitate a clearer understanding of our experimental models, we additionally provide illustrations of (a), (b), and (c) in Fig. B of the PDF in the global response.
>
> ||Model|mAP@0.3|mAP@0.5|mAP@0.7|mAP@Avg|
> |-|-|-|-|-|-|
> |(a)|ActionFormer w/o cross-modal fusion|30.3|15.1|3.5|16.1|
> |(b)|Cross-modal baseline (self-attn)|42.2|24.0|5.5|23.8|
> |(c)|**Cross-modal baseline (cross-attn)**|**55.9**|**35.9**|**10.5**|**34.5**|
>
> This table shows that ActionFormer without cross-modal fusion (a) exhibits inferior performance because text and video are not properly aligned, highlighting the importance of cross-modal fusion.
> Furthermore, the cross-attention baseline (c) outperforms the self-attention baseline (b).
> We will add these results and illustrations to the Appendix to emphasize the contributions of our cross-modal part.
>
> > **W3) The authors only use I3D visual feature for experiments in Table 1, it would be help to evaluate with CLIP features to demonstrate the generalization ability to different features.**
>
> We add experimental results utilizing the CLIP as a visual encoder in Table 1.
> We have revised Table 1, including an experiment utilizing a CLIP visual encoder.
> Please refer to Table 1 in the PDF of the global response.
>
> As shown in the revised Table 1, the performance using I3D features still outperforms the use of CLIP features because the simultaneous utilization of spatiotemporal information is crucial in the temporal action detection area. Therefore, in the original manuscript, we primarily utilize I3D as a visual encoder, following previous works [2,3,6]. However, we agree that these experiments utilizing CLIP as a visual encoder are essential to demonstrate the generalization ability of our method with different features.
>
> > **W4) Limitations are not discussed in the paper.**
>
> We have mentioned some limitations in Sec. 5 on lines 302-304.
> However, we acknowledge the need for a more detailed discussion.
> We will extend the limitation part.

---

> > ### Comment · Reviewer_mst5 · 2024-08-13
> >
> > Thanks for providing the detailed response and conducting additional experimental results. My concerns have been well-addressed, and I would like to keep my initial rating as weak accept.

---

### Official Review · Reviewer_QEJb · 2024-07-13

**Soundness:** 3
**Presentation:** 3
**Contribution:** 3
**Rating:** 5
**Confidence:** 5

**Summary:**

This paper deals with the problem of Zero-Shot Temporal Action Detection (ZSTAD). The authors propose a simple cross-modal ZSTAD baseline with good performance. To address the issue that the cross-modal baseline over-focus on common sub-actions, the paper further proposed a Ti-FAD module to focus on text-related visual parts. The method is evaluated on two popular datasets.

**Strengths:**

1. The paper is generally well-written and easy to follow. The proposed method is well presented.
2. The issue that the cross-modal baseline over-focus on common sub-actions is well recognized and analyzed. The proposed Ti-FAD module seems to be effective on this issue.
3. The performance of the proposed method is promising.

**Weaknesses:**

1. The two branches in Figure 2(a) and Figure 3(a) should be cross-connected before cross-attention. Now they look like they are independent.

2. The TiCA module seems a bit counter-intuitive. After obtaining the Smask, which describes which visual parts are most text-related, you didn't use it as an attention mask for cross-attention from text to vision, but instead, as a bias for vision-to-text attention. What was the rationale behind this decision? It might seem more intuitive to use the latter operation, where the Smask masks out unimportant visual segments, allowing the text to focus more on the important visual fragments. Besides, I would like to see a comparison in the experiment.

3. The so-called "Foreground-Aware" approach refers to the model's ability to predict temporal boundaries on its own, without relying on an offline proposal generator, is that right? The design in the Foreground-Aware Head is quite common in temporal action detection methods, and I don't see anything particularly unique about it. Is this a core contribution of the article?

**Questions:**

See weakness.

**Limitations:**

The authors state limitations in the paper.

---

> ### Author Rebuttal · Authors · 2024-08-06
>
> > **W1) The two branches in Fig. 2(a) and 3(a) should be cross-connected before cross-attention.**
>
> We apologize for any potential misleading caused by Fig. 2(a) and 3(a).
> As shown in Eq. (1) and (6), ${F'}^{(l)}\_{vid}$ and ${F'}^{(l)}\_{txt}$ are used as the inputs for the cross-attention, and these notations are also depicted inside of X-MHA and TiCA in Fig. 2(a) and 3(a).
> To enhance the clarity of our method, we have added cross-connected arrows to both figures.
> The updated figures are included in the PDF of the global response.
> We hope that adding the cross-connected arrows can clarify our method.
>
>
> > **W2) The TiCA module seems a bit counter-intuitive. After obtaining the $S_{mask}$, which describes which visual parts are most text-related, you didn't use it as an attention mask for cross-attention from text to vision, but instead, as a bias for vision-to-text attention. What was the rationale behind this decision? It might seem more intuitive to use the latter operation, where the $S_{mask}$ masks out unimportant visual segments, allowing the text to focus more on the important visual fragments. Besides, I would like to see a comparison in the experiment.**
>
> In this paper, our main goal is to ensure that the cross-attention part utilizes text-related discriminative visual features, regardless of the cross-attention direction.
> We primarily focus on how to obtain the discriminative part using the salient attentive mask (SAM, $S_{mask}$) to address text-related discriminative visual features by containing information about which temporal location is more text-related.
> Thereby, the direction of the cross-attention does not change the meaning of the SAM.
> Furthermore, we believe that a video-centric approach is more beneficial than a text-centric approach because zero-shot TAD involves more extensive information from video than text (that just contains the names of actions).
> The rationale behind this decision aligns with our experimental results comparing three approaches: using $S_{mask}$ as an attention bias for text-to-vision cross-attention, vision-to-text cross-attention, and both directions of cross-attention.
> To apply text-to-vision cross-attention, we transpose $S\_{mask} \otimes \mathbf{1}\_{C}$ in Eq. (6).
> The performance comparison is shown in the tables below.
>
> **50%-50% setting**
> |Method|THUMOS14 mAP@0.3|mAP@0.5|mAP@0.7|mAP@Avg|ActivityNet v1.3 mAP@0.5|mAP@0.75|mAP@0.95|mAP@Avg|
> |-|-|-|-|-|-|-|-|-|
> |Baseline|55.9|35.9|10.5|34.5|44.1|26.1|3.1|26.6|
> |Text-to-vision|**57.4**|42.5|20.0|40.6|50.8|31.9|**5.6**|32.0|
> |Vision-to-text (Ours)|57.0|**43.3**|**21.2**|**41.2**|50.6|**32.2**|5.2|32.0|
> |Both|56.2|42.8|**21.2**|40.7|**50.9**|32.1|**5.6**|**32.1**|
>
> **75%-25% setting**
> |Method|THUMOS14 mAP@0.3|mAP@0.5|mAP@0.7|mAP@Avg|ActivityNet v1.3 mAP@0.5|mAP@0.75|mAP@0.95|mAP@Avg|
> |-|-|-|-|-|-|-|-|-|
> |Baseline|59.1|39.3|12.2|37.4|47.0|27.2|3.8|28.3|
> |Text-to-vision|62.3|47.8|23.6|45.4|**53.9**|34.5|6.3|34.3|
> |Vision-to-text (Ours)|**64.0**|**49.7**|24.1|**46.8**|53.8|**34.8**|**7.0**|**34.7**|
> |Both|63.6|48.8|**24.5**|46.5|**53.9**|34.4|6.1|34.2|
>
> As shown in the table, utilizing video features as a query (vision-to-text) slightly outperforms compared to other methods.
> As you mentioned, text-to-vision also allows the text features to focus on the related visual features and significantly improves the performance compared to our baseline.
> Our model shows similar performance regardless of whether $S_{mask}$ is used for text-to-vision or vision-to-text cross-attention. These results indicate that the most crucial part of our TiCA is the salient attentive mask (SAM, $S_{mask}$), and the choice between text-to-vision and vision-to-text cross-attention is less critical.
> We will revise the Ti-FAD section to emphasize the focus on $S_{mask}$ more clearly.
>
>
> > **W3) The so-called "Foreground-Aware" approach refers to the model's ability to predict temporal boundaries on its own, without relying on an offline proposal generator, is that right?**
>
> As you mentioned, our model does not require an offline proposal generator. However, the "Foreground-Aware" approach is not related to predicting temporal boundaries. As described in Lines 173-174, the "Foreground-Aware head" aims for our model to focus more on foreground action segments in a class-agnostic manner.
>
>
> > **W3) The design in the Foreground-Aware Head is quite common in temporal action detection methods, and I don't see anything particularly unique about it. Is this a core contribution of the article?**
>
> Our core novelties are conducting solid cross-modal baseline, as acknowledged by Reviewer EXtU, and proposing the Ti-FAD that addresses the issue of over-focus on common sub-actions.
> In this context, we mainly focus on capturing text-related sub-action parts using the salient attentive mask (SAM, $S_{mask}$) in TiCA and reducing the influence of unrelated parts by suppressing the background in the foreground-aware head.
> Therefore, our contribution not only lies in adopting the foreground-aware head in our Ti-FAD.
> Furthermore, while some aspects of the foreground-aware head are similar to existing methods, utilizing foreground-awareness is first applied in the zero-shot temporal action detection area.

---

> > ### Comment · Reviewer_QEJb · 2024-08-12
> >
> > Thanks to the authors for their detailed rebuttal.
> >
> > My concerns have been addressed. Considering the overall quality of the paper and the comments of other reviewers, I keep my rating at "Borderline accept".

---

### Official Review · Reviewer_EXtU · 2024-07-18

**Soundness:** 2
**Presentation:** 2
**Contribution:** 3
**Rating:** 5
**Confidence:** 4

**Summary:**

The paper provides a solution for zero-shot temporal action detection(TAD). It mainly addresses distinguishing between similar actions that share common sub-actions. The method introduces a cross-attention in the model and two actionness losses. They benchmark on the two standard TAD datasets, Thumos and ActivityNET, and outperform existing methods by a significant margin.

**Strengths:**

1) The proposed method outperforms existing methods by a significant margin
2) The paper mentions an interesting issue of the classes in the dataset containing a common sub-action, which could confuse the model.
3) The paper addresses a challenging and relatively underexplored task of zero-shot TAD.

**Weaknesses:**

1) From my understanding, none of the previous methods deployed the ActionFormer baseline. This is a robust baseline, as you mentioned in L75. You only report the results of the cross-modal baseline, claiming that introducing the cross-modal model boosts the performance over the existing models. However, I don't see the results of using the ActionFormer detector without the baseline modifications. I think that would give a more fair view of the contributions of the cross-modal part.

2) Your first contribution states the novelty of the cross-modal baseline. From what I understand from Sec.3.2, you are introducing a well-established cross-attention module into ActionFormer. That is also important to consider in 1).

3) How often does the baseline model need clarification due to a common action bias? For example, if you plot the confusion matrix, would you actually see that it most often predicts the wrong class containing the common action? Furthermore, do you think this issue is specific to datasets like Thumos, where the granularity of actions is not that fine?

4) [L70-71] If you are using CLIP, you must provide a specific text (prompt) for this model, even if that is just a class name. Some models, such as UnLOC, just had one prompt to provide the results of the zero-shot, which augments the class name with a prefix text. This does not seem so different from what you do.

5) Please clarify more explicitly what SAM stands for (I assume it should be the salient attentive mask). The first time it is used in the text is in Figure 3, and it is already an abbreviation.

6) [Table 1] Does UNLoc do prompt tuning? I don't see any mention in the paper of any learnable prompts. You also report the results of UNLoc using a simple single prompt only, so I would not say that they are doing much engineering on the prompt side for these results.

**Questions:**

Why are you using the I3D features and not the CLIP vision encoder features?

Also, since you don't fine-tune the text encoder, did you try using a larger CLIP-L? The results in UNLoc with a larger CLIP-L are better than with CLIP-B. I am curious to know if that would hold for you, given that you only use CLIP for the text part (I am not asking for an experiment on this one, just in case you have run it already).

Overall, I think the results on Thumos and ANET are good, but I am concerned about why you didn't include the ActionFormer baseline. I am willing to change my opinion if my points are addressed.

**Limitations:**

The authors briefly mentioned the limitations in the conclusion.

---

> ### Author Rebuttal · Authors · 2024-08-06
>
> > **W1,W2) ActionFormer detector without cross-modal modification. I think that would give a more fair view of the contributions of the cross-modal part.**
>
> We appreciate your in-depth comment, which allows our cross-modal part to be viewed more fairly.
> We conduct a comparative analysis of ActionFormer without the cross-modal fusion and our cross-modal baselines on THUMOS14 in the 50%-50% setting.
> To facilitate a clearer understanding of our experimental models, we additionally provide illustrations of (a), (b), and (c) in Fig. B of the PDF in the global response.
>
> ||Model|mAP@0.3|mAP@0.5|mAP@0.7|mAP@Avg|
> |-|-|-|-|-|-|
> |(a)|ActionFormer w/o cross-modal fusion|30.3|15.1|3.5|16.1|
> |(b)|Cross-modal baseline (self-attn)|42.2|24.0|5.5|23.8|
> |(c)|**Cross-modal baseline (cross-attn)**|**55.9**|**35.9**|**10.5**|**34.5**|
>
> This table shows that ActionFormer without cross-modal fusion (a) exhibits inferior performance because text and video are not properly aligned, highlighting the importance of cross-modal fusion.
> Furthermore, the cross-attention baseline (c) outperforms the self-attention baseline (b).
> We will add these results and illustrations to the Appendix to provide a fairer view of our cross-modal part.
>
> > **W3) How often does the baseline need clarification due to a common action bias? If you plot the confusion matrix, would you see that it most often predicts the wrong class containing the common action?**
>
> Unlike classification tasks, zero-shot TAD involves detecting and localizing actions within continuous video streams.
> This makes it challenging to visualize misclassifications in the form of a confusion matrix.
> Instead, we utilize Average Precision (AP) following the previous work [1], which provides a comprehensive measure of performance in detecting and classifying actions.
> To show which action class is mostly affected by common sub-action bias, we compared the per-unseen class AP on THUMOS14 in Fig. 4 (Main paper) and A (Appendix).
> By comparing AP between our baseline and Ti-FAD, we show that our Ti-FAD successfully addresses the common sub-action bias by improving the AP for action classes such as *LongJump*, *HighJump*, *PoleVault*, *JavelinThrow*, and *BasketballDunk* that share the common sub-action part *Running*.
> These improvements are not only reflected in the performance metrics but also align with our expectations.
>
> > **W3) Do you think this issue is specific to datasets like Thumos, where the granularity of actions is not that fine?**
>
> We observe that sharing common sub-actions between multiple classes also occurs in other datasets, such as ActivityNet v1.3.
> For example, actions such as *Painting*,  *Cleaning sink*, *Washing face*, *Cleaning windows*, *Hand car wash*, *Cleaning shoes*, *Ironing clothes*, *Hand washing clothes*, and *Washing dishes* that share the common sub-action part *Wiping motion with hands*, leading to similar issue.
> Furthermore, we provide additional per-unseen class AP results for 20 classes of ActivityNet v1.3 compared with our cross-modal baseline results in the PDF of the global response.
> The AP results for all classes will be added to the Appendix.
> These results demonstrate that the common sub-action bias issue is not isolated to THUMOS14.
>
> > **W4) You must provide a specific text prompt. Some models, such as UnLoc, just had one prompt, which augments the class name with a prefix text.**
>
> We simply use "{classname}" as the text prompt without any prefix or contextual text.
> We will add a detailed description of the text prompt in Line 107 to provide a clear view.
> Furthermore, to show that Ti-FAD's performance does not depend on the type of text prompt, we compare the performance of our baseline and Ti-FAD with the most utilized types of text prompts following previous works [2,3,4,6] on THUMOS14 in the 50%-50% setting.
>
> ||Text Prompt|Baseline mAP@0.3|mAP@0.5|mAP@0.7|mAP@Avg|Ti-FAD mAP@0.3|mAP@0.5|mAP@0.7|mAP@Avg|
> |-|-|-|-|-|-|-|-|-|-|
> |(a)|"a video of action {classname}"|53.8|33.9|9.7|32.8|55.6|42.6|20.9|40.4|
> |(b)|Prompt Augmentation [5]|54.3|34.6|10.3|33.4|56.3|42.7|20.6|40.5|
> |(c)|Prompt Ensemble [5]|53.8|34.3|10.2|33.1|56.8|43.0|20.3|40.7|
> |(d)|**"{classname}"**|**55.9**|**35.9**|**10.5**|**34.5**|**57.0**|**43.3**|**21.2**|**41.2**|
>
> (a) is used in [2,3,6].
> (b) refers to using the 28 templates of the prompt, used in UnLoc [4].
> (c) refers to using the average embedding vector from the 28 templates at the inference, used in UnLoc [4].
> The result demonstrates that Ti-FAD shows similar performance regardless of the type of text prompt.
> We will add this experiment to the Appendix.
>
> > **W5) Clarifying the term of SAM.**
>
> We apologize for any confusion caused by the unclear term. We will replace "SAM" with "salient attentive mask (SAM)" in Line 158, which is used for the first time.
>
> > **W6) Does UNLoc do prompt tuning?**
>
> We apologize for any confusion in Table 1. As you mentioned, UnLoc [4] does not use prompt tuning.
> We have revised Table 1, including UnLoc's all reported versions.
> Please check Table 1 in the PDF of the global response.
>
> > **Q1) Why are you using the I3D features and not the CLIP features?**
>
> In video understanding tasks [2,3,6], using snippet-level features is generally more effective than frame-level features for capturing temporal details because addressing simultaneous spatio-temporal information is essential for video understanding.
> For a fair comparison, we add the results using the CLIP features in Table 1.
> The revised Table 1 shows that I3D features still outperform overall.
> Even when using CLIP as a video encoder, our Ti-FAD still shows competitive performance compared to previous methods using I3D features.
> Please check Table 1 in the PDF of the global response.
>
> > **Q2) Did you try using a CLIP-L?**
>
> Following the most existing methods [2,3,6], we mainly utilize CLIP-B. For a more comprehensive comparison, we have updated the results using CLIP-L in Table 1.
> Please check Table 1 in the PDF of the global response.

---

> ### Comment · Reviewer_EXtU · 2024-08-12
>
> Dear authors,
>
> I appreciate all your answers!
> I have a short follow-up question: Why does using CLIP-L versus CLIP-B decreases the performance for your model?

---

> > ### Author Response · Authors · 2024-08-13
> >
> > We appreciate your constructive feedback once again and your additional question, which has allowed us to clarify our findings.
> >
> > We also want to express our gratitude for the time and effort you have dedicated to reviewing our work.
> >
> > ---
> >
> > Based on our analysis, it appears that there is no significant difference in performance between CLIP-B and CLIP-L as text encoders in most cases.
> > We have split our findings into two scenarios in the context of video encoder (CLIP, I3D):
> >
> > **Video encoder: CLIP**
> >
> > As shown in the revised Table 1, using CLIP-L as a text encoder performs worse than using CLIP-B at the 50%-50% setting on THUMOS14.
> > However, using CLIP-L as a text encoder shows superior performance compared to CLIP-B at the 75%-25% setting on THUMOS14 and both settings (50%-50% and 75%-25%) on ActivityNet v1.3.
> >
> > **Video encoder: I3D**
> >
> > As shown in the revised Table 1, using CLIP-L as a text encoder shows worse performance compared to CLIP-B at the 50%-50% setting on THUMOS14 and 75-25% setting on ActivityNet v1.3.
> > However, using CLIP-L as a text encoder shows superior performance compared to CLIP-B at the 75%-25% setting on THUMOS14 and 50-50% setting on ActivityNet v1.3.
> >
> > In conclusion, this mixed performance indicates that the choice of text encoder does not significantly impact performance because **zero-shot temporal action detection involves more extensive information from video than text (just contains the names of actions)**.
> >
> > ---
> >
> > We hope that our overall rebuttal and this note have addressed your concerns thoroughly.
> >
> > We would like to carefully request your consideration in increasing the final review score.

---

> > > ### Comment · Reviewer_EXtU · 2024-08-14
> > >
> > > I want to thank authors for their response.
> > >
> > > I am increasing my rating to borderline accept.

---

### Author Rebuttal · Authors · 2024-08-06

Dear All Reviewers,

We sincerely appreciate the reviewers' thoughtful feedback on our paper.
We are grateful for the time and effort you have taken to review our manuscript.
All constructive comments allowed us to develop our paper even further.


In this global response, we address three aspects:
1. **Our Core Novelties**
2. **Key Strengths Acknowledged by Reviewers**
3. **Additional Experiments & Revised Table and Figures (PDF)**

---

### **Our Core Novelties**

1. **Novel Cross-Modal Baseline**:
   - We introduce a solid cross-modal Zero-Shot Temporal Action Detection (ZSTAD) baseline, which shows strong performance compared to the previous methods. Our cross-modal baseline enables the detecor to capture the whole information of both video and text modalities throughout the entire detection process by eliminating the step of pre-extracting foreground candidate proposals in the previous ZSTAD methods.

2. **Identifying Key Issue**:
   - Our approach identifies the common sub-action bias issue causing confusion in our cross-modal baseline. For example, in the THUMOS14 case, action classes such as *LongJump*, *HighJump*, *PoleVault*, *JavelinThrow*, and *BasketballDunk* can share the common sub-action part *Running*.  In the ActivityNet v1.3 case, action classes such as *Painting*,  *Cleaning sink*, *Washing face*, *Cleaning windows*, *Hand car wash*, *Cleaning shoes*, *Ironing clothes*, *Hand washing clothes*, and *Washing dishes* can share the common sub-action part *Wiping motion with hands*. These common sub-actions hinder the model from learning the fine-grained alignment between video and text information, especially in a ZSTAD environment where the amount of video information is overwhelmingly larger than the text information (that just contains the name of action).

3. **Model Innovation**:
   - The proposed Text-infused attention and Foreground-aware Action Detection (Ti-FAD) addresses the common sub-action bias issue by (1) capturing text-related sub-action parts, and (2) distinguishing action segments from the background.
Our Ti-FAD shows promising performance on THUMOS14 and ActivityNet v1.3.

---

### **Key Strengths Acknowledged by Reviewers**

We are encouraged by the reviewers' recognition of our approach in several key areas:
1. **Outperforming Existing Methods** (Reviewers EXtU, QEJb, and mst5):
   - Our method significantly outperforms existing methods by a significant margin.

2. **Addressing the Important Issue about Common Sub-Action Confusion** (Reviewers EXtU, QEJb, and mst5):
   - Our paper identifies and addresses the profound issue of sub-actions causing confusion in our baseline.

3. **Clarity and Readability** (Reviewers QEJb and mst5):
   - Our paper is well-written and easy to follow.

4. **Tackling an Unexplored and Challenging Task** (Reviewer EXtU):
   - Our paper addresses an unexplored and challenging zero-shot approach in the temporal action detection area.

---

### **Additional Experiments & Revised Table and Figures (PDF)**

1. **Table 1**: The revised version of Table 1 in the original manuscript. This table additionally contains three aspects: (1) Additional experiments for CLIP visual encoders, (2) Additional experiments for CLIP-L text encoder, and (3) missing part of a previous work.

2. **Figure 2(a) and Figure 3(a)** : Updated model figures for improved clarity. We have added cross-connected arrows to depict the connection of cross-attention.

3. **Figure B**: Illustration of the baseline approach to clarify the experimental setup in Reviewers EXtU's comment (W1,W2) and mst5's comment (W2,Q2). This figure demonstrates the differences in baseline architectures of the ActionFormer model without cross-modal fusion and cross-modal baselines (self-attention and cross-attention). Our response to Reviewers EXtU's comment (W1,W2) and mst5's comment (W2,Q2) contains the comparative experimental results and discussion about this part.

4. **Figure C**: Per-unseen class AP for 20 classes in ActivityNet v1.3. This figure shows the improvements of our Ti-FAD compared to our cross-modal baseline, which indicates that our Ti-FAD addresses common sub-action confusion in various datasets.

---
**References**

[1] L. Zhang et al., "ZSTAD: Zero-Shot Temporal Activity Detection," *CVPR*, 2020.

[2] S. Nag et al., "Zero-Shot Temporal Action Detection via Vision-Language Prompting," *ECCV*, 2022.

[3] C. Ju et al., "Prompting Visual-Language Models for Efficient Video Understanding," *ECCV*, 2022.

[4] S. Yan et al., "UnLoc: A Unified Framework for Video Localization Tasks," *ICCV*, 2023

[5] A. Radford et al., "Learning Transferable Visual Models from Natural Language Supervision," *ICML*, 2021.

[6] T. Phan et al., "ZEETAD: Adapting Pretrained Vision-Language Model for Zero-Shot End-to-End Temporal Action Detection," *WACV*, 2024.

---

We sincerely appreciate all constructive comments and believe that the revisions and additional experiments have strengthened our paper.

We are also committed to making our code publicly available upon acceptance of the paper, to facilitate further research in this area.

We kindly request your consideration for a positive evaluation.

Once again, we appreciate all your valuable feedback.

---

### Decision · Program_Chairs · 2024-09-25

**Decision:**

Accept (poster)

**Comment:**

This paper presents a novel approach for Zero-Shot Temporal Action Detection (ZSTAD), which introduces a cross-modal baseline that integrates both text and visual features throughout the detection process. The paper received three reviews with unanimous recommendations of accepting the paper, mainly due to its technical novelty, strong experimental justification, clear presentation, and comprehensive experimental evaluation. The concerns raised by reviewers, especially regarding the baseline comparisons and design choices, were effectively addressed in the rebuttal. Thus, the AC suggests to accept the paper. The authors are suggested to consider the review comments to further refine the paper for preparing the final version, especially in the presentation and justification of design decisions.